# Targeting Oncogenic Mutant p53 and BCL-2 for Small Cell Lung Cancer Treatment

**DOI:** 10.3390/ijms241713082

**Published:** 2023-08-23

**Authors:** Victoria Neely, Alekhya Manchikalapudi, Khanh Nguyen, Krista Dalton, Bin Hu, Jennifer E. Koblinski, Anthony C. Faber, Sumitra Deb, Hisashi Harada

**Affiliations:** 1Philips Institute for Oral Health Research, School of Dentistry, Massey Cancer Center, Virginia Commonwealth University, Richmond, VA 23298, USA; neelyvl@vcu.edu (V.N.); alekmanchi@gmail.com (A.M.); nguyenkt22@vcu.edu (K.N.); powellkm5@vcu.edu (K.D.); acfaber@vcu.edu (A.C.F.); 2Department of Pathology, School of Medicine, Massey Cancer Center, Virginia Commonwealth University, Richmond, VA 23298, USA; bhu@vcu.edu (B.H.); jennifer.koblinski@vcuhealth.org (J.E.K.); 3Department of Biochemistry & Molecular Biology, School of Medicine, Massey Cancer Center, Virginia Commonwealth University, Richmond, VA 23298, USA; sumitra.deb@vcuhealth.org

**Keywords:** small cell lung cancer, p53, BCL-2, BIM, venetoclax, HSP90, ganetespib, targeted therapy

## Abstract

Through a unique genomics and drug screening platform with ~800 solid tumor cell lines, we have found a subset of SCLC cell lines are hypersensitive to venetoclax, an FDA-approved inhibitor of BCL-2. SCLC-A (ASCL1 positive) and SCLC-P (POU2F3 positive), which make up almost 80% of SCLC, frequently express high levels of BCL-2. We found that a subset of SCLC-A and SCLC-P showed high BCL-2 expression but were venetoclax-resistant. In addition, most of these SCLC cell lines have TP53 missense mutations, which make a single amino acid change. These mutants not only lose wild-type (WT) p53 tumor suppressor functions, but also acquire novel cancer-promoting activities (oncogenic, gain-of-function). A recent study with oncogenic mutant (Onc)-p53 knock-in mouse models of SCLC suggests gain-of-function activity can attenuate chemotherapeutic efficacy. Based on these observations, we hypothesize that Onc-p53 confers venetoclax resistance and that simultaneous inhibition of BCL-2 and Onc-p53 induces synergistic anticancer activity in a subset of SCLC-A and SCLC-P. We show here that (1) down-regulation of Onc-p53 increases the expression of a BH3-only pro-apoptotic BIM and sensitizes to venetoclax in SCLC-P cells; (2) targeting Onc-p53 by the HSP90 inhibitor, ganetespib, increases BIM expression and sensitizes to venetoclax in SCLC-P and SCLC-A cells. Although there are currently many combination studies for venetoclax proposed, the concept of simultaneous targeting of BCL-2 and Onc-p53 by the combination of venetoclax and HSP90 inhibitors would be a promising approach for SCLC treatment.

## 1. Introduction

Small-cell lung cancer (SCLC) is a high-grade neuroendocrine (NE) carcinoma that accounts for 15~20% of all lung cancers and is commonly associated with a significant tobacco history. Patient outcomes have not improved substantially over the last 30 years, underscoring the need for more effective treatment strategies [1,2]. Despite initial sensitivity to platinum-based combination chemotherapies, SCLCs characteristically develop progressive chemotherapy resistance, resulting in a five-year survival rate of approximately 5%. The genomic analyses with SCLCs demonstrate a predictable pattern of mutations of tumor suppressors *RB1* and *TP53* (in almost all cases), loss of *PTEN* or *PIK3CA* mutations (in up to 20% of cases), mutations in *CREBBP*/*EP300* histone acetyltransferases (in ~15% of cases), and *MYC* transcription factor family amplification (in ~25% of cases) [3,4,5]. In addition, the pro-survival BCL-2 is highly expressed in 65% of cases [6].

The recent addition of pro-survival BCL-2 family inhibitors (BH3 mimetics) to the repertoire of targeted therapies revealed marked pre-clinical activity in SCLC [7]. Unfortunately, in a phase II trial with 39 SCLC patients using the dual BCL-2 and BCL-X_L_ inhibitor, navitoclax (ABT-263), only 1 patient achieved a partial response and 8 patients had stable disease [8]. While the response rate was disappointing, the disease control rate of 26% in a refractory population suggests some single-agent activity. Importantly, it was noted that due to the BCL-X_L_ inhibition [9], there was dose-limiting thrombocytopenia [8,10]. This strongly suggests that the magnitude of BCL-2 inhibition was limited in this trial by the toxicities induced by the BCL-X_L_ targeting of navitoclax, and treatment with a BCL-2 specific inhibitor would be expected to avoid thrombocytopenia and therefore more potently inhibit BCL-2. Indeed, venetoclax (ABT-199), an FDA-approved BCL-2-specific inhibitor, does just this, and has revolutionized treatment for hematologic cancers [11,12]. Now venetoclax provides, for the first time, a drug to test the hypothesis that a subset of SCLCs are dependent on BCL-2 and may respond to pharmaceutical inhibition of BCL-2. In fact, our recent data provide evidence that BCL-2 is a promising drug target for high BCL-2-expressing SCLCs [13]. 

While SCLC with variable NE marker expression has been appreciated for 35 years, the leading experts in SCLC biology have recently proposed a classification of four SCLC subtypes based on the expression of key transcription factors: SCLC-A (ASCL1 positive), SCLC-N (NeuroD1 positive), SCLC-P (POU2F3 positive), and SCLC-Y (YAP1 positive). Among them, SCLC-A and SCLC-P, which make up almost 80% of SCLC, frequently express high levels of BCL-2 [14]. We noticed that the BCL-2-high and venetoclax-sensitive cell lines that we examined all belong to SCLC-A. However, we also found that a subset of SCLC-A and SCLC-P showed high BCL-2 expression, but were venetoclax-resistant. In addition, most of these SCLC cell lines have *TP53* missense mutations, which make a single amino acid change. These mutants not only lose wild-type (WT) p53 tumor suppressor functions, but also acquire novel cancer-promoting activities (oncogenic, gain-of-function) [15,16,17]. It has been shown that Onc-p53 expression is associated with chemo- and radio-resistance in a variety of cancers [18,19,20]. Furthermore, a recent study with Onc-p53 knock-in mouse models of SCLC suggests Onc-p53 activity can attenuate chemotherapeutic efficacy [21]. Based on these observations, we hypothesize that Onc-p53 confers venetoclax resistance and that simultaneous inhibition of BCL-2 and Onc-p53 induces synergistic anticancer activity in a subset of SCLC-A and SCLC-P. Here, we show that (1) down-regulation of Onc-p53 increases the expression of a pro-apoptotic BH3-only BCL-2 family protein, BIM, and sensitizes to venetoclax in SCLC-P cells; (2) targeting Onc-p53 by the HSP90 inhibitor, ganetespib [22], increases BIM expression and sensitizes to venetoclax in SCLC-P and SCLC-A cells. We further investigated the mechanism of action of this combination in vitro and the efficacy of this combination in a mouse xenograft model of SCLC. Collectively, the study provides us with a proof of principle of the efficacy of HSP90 inhibitors in combination with the BCL-2 inhibitor for SCLC treatment.

## 2. Results

### 2.1. Down-Regulation of Onc-p53 Increases BIM Expression and Sensitizes to Venetoclax in SCLC-P Cells

In order to test our hypothesis that simultaneous inhibition of BCL-2 and Onc-p53 induces synergistic anticancer activity in a subset of SCLC-A and SCLC-P, we chose two BCL-2 high SCLC-P cell lines, H1048 and H211 to down-regulate Onc-p53 (R273C in H1048 and R248Q in H211) by shRNA (scrambled shRNA was used as a control). First, we determined the levels of the BCL-2 family proteins in these cells (Figure 1A and Appendix A). When Onc-p53 was downregulated, the levels of pro-survival BCL-2, BCL-X_L_, and MCL-1 were not significantly altered. Interestingly, a pro-apoptotic BH3-only protein, BIM, was increased by down-regulation of Onc-p53 in both cell lines, suggesting the cells are in a state of “prime to death” [23]. The increased level of BIM protein correlated with the increase in mRNA (Figure 1B and Appendix A), suggesting that Onc-p53 directly or indirectly suppresses BIM transcription. Since we have shown that BIM is critical for venetoclax-induced apoptosis in BCL-2-high SCLC cells [13], the result prompted us to examine whether venetoclax-resistant H1048 and H211 cells could be sensitized by down-regulation of Onc-p53. Importantly, the concentration used in these assays is clinically relevant based on steady-state pharmacokinetics observed with venetoclax [24,25]. Venetoclax-treated H1048 shp53 cells showed faster (6 h) and more prominent (16 h) cleaved-caspase 3 expression compared to control cells (Figure 1C). Similarly, H211 shp53 cells showed more cleaved-caspase 3 expression compared to control cells (Appendix A), indicating that inhibition of Onc-p53 sensitizes to venetoclax. Taken together, these results suggest that Onc-p53, at least in part, contributes to venetoclax resistance in BCL-2-high SCLC-P cells. 

### 2.2. Targeting Onc-p53 by the HSP90 Inhibitor, Ganetespib, Increases BIM Expression and Sensitizes to Venetoclax in SCLC Cell Lines

One characteristic of Onc-p53 is protein stabilization compared to WT p53. This stabilization is one key pre-requisite for Onc-p53 and is largely due to mutant p53 protection from the E3 ubiquitin ligases Mdm2 and CHIP by the HSP90/HDAC6 chaperone machinery [16]. It has been demonstrated that inhibition of HSP90 alone with ganetespib has marked anti-tumoral effects in vivo and extends the overall survival of p53-R175H and p53-R248Q knock-in mice [26]. Notably, p53-null mice did not benefit from HSP90 inhibition. These anticancer effects are concomitant with mutant p53 degradation and cancer cell death, indicating tumor addiction to stabilized mutant p53 [26]. Based on these observations, we first examined whether ganetespib treatment can induce degradation of Onc-p53 in SCLC-P cells. p53-R273C in H1048 was degraded by ganetespib in a dose-dependent manner (Figure 2A). Interestingly, BIM protein and mRNA were up-regulated in accordance with Onc-p53 degradation, which could be reproduced by knock-down of Onc-p53 (Figure 2A,B). We next treated the cells with a combination of venetoclax and ganetespib. The combination induced more cell death compared with ganetespib alone in H1048 cells (Figure 2C). We confirmed that the combination induced more cleaved caspase 3, indicative of apoptosis, compared with single drug treatments (Figure 2D). 

The enhancement of cell death by combination treatment was also observed in SCLC-P H211 cells as well as venetoclax-resistant SCLC-P H526 cells, both of which have mutant p53 (Figure 3A,B). In addition, venetoclax-resistant and mutant p53-expressing SCLC-A H345 and H146 cells also show enhancement of cell death in combination (Figure 3C,D). However, the combination-induced cell death was not observed in SCLC-A H209 cells with WT p53 (Figure 4A,B), indicating that the effect of the HSP90 inhibitor is specific for Onc-p53 (proof of concept). Furthermore, another clinically tested HSP90 inhibitor, XL888 [27], also shows enhancement of cell death by combination with venetoclax (Figure 4C,D), suggesting general effects of HSP90 inhibitors in the proposed combination treatment. Taken together, these data support our hypothesis that Onc-p53 confers venetoclax resistance and that simultaneous inhibition of BCL-2 and Onc-p53 enhances anticancer activity in a subset of SCLC-A and SCLC-P.

### 2.3. A Pro-Apoptotic BH3-Only Protein, BIM, Is Required for Cell Death in Combination of Venetoclax and Ganetespib Treatment in SCLC Cells

We have previously demonstrated that the disruption of the BIM:BCL-2 complex by venetoclax leads to BIM-dependent apoptosis in venetoclax-sensitive SCLC cells [13]. Thus, we first investigated the significance of BIM in cell death with the venetoclax and ganetespib combination. Down-regulation of BIM by shRNA in H1048 and H211 cells mitigated cell death by venetoclax and ganetespib combination as determined by cell viability, Western blots, and apoptosis analysis by FACS (Figure 5 and Appendix A). 

Venetoclax is known to bind to the hydrophobic pocket of BCL-2 to inhibit the formation of the BIM:BCL-2 complex, followed by the activation of BAX, the downstream effector of BIM, to induce apoptosis [11,13]. We performed co-immunoprecipitation experiments to determine whether the amount of BIM:BCL-2 complex was affected by venetoclax treatment. The bands detected by immunoprecipitation in H1048 whole-cell lysates were specific to anti-BIM or anti-BCL-2 antibodies, as control IgG antibodies did not show non-specific bands (Appendix A). Immunoprecipitation of BIM in H1048 whole-cell lysates treated with venetoclax and ganetespib revealed on-target activity of venetoclax as judged by the disassociation of BIM:BCL-2 complexes, comparing ganetespib alone (Lane 2 in Figure 6A,B) with ganetespib + venetoclax (Lane 3 in Figure 6A,B). As a consequence, BAX was conformationally changed and activated as more immunoprecipitated by the conformation-specific antibody, 6A7 (Lane 3 in Figure 6C, [28]). Taken together, a BH3-only protein, BIM, plays a critical role in venetoclax and ganetespib treatment in SCLC cells expressing Onc-p53.

### 2.4. The Combination of Venetoclax and Ganetespib Treatment Delays Tumor Growth in a SCLC Xenograft Mouse Model

Finally, we tested the efficacy of the venetoclax and ganetespib combination in a SCLC xenograft. H1048 cells were subcutaneously inoculated at the flank, and once tumors became pulpable, mice were randomized into four groups treated with venetoclax and ganetespib alone or in combination. Either venetoclax or ganetespib alone showed modest reductions in tumor volume compared with vehicle injection, whereas combined exposure to venetoclax and ganetespib significantly reduced tumor volume at the experimental endpoint by ~50% (Figure 7A). During the treatment period, there was no sign of weight loss (Figure 7B), suggesting that the combination treatment would be well tolerated. All results encouraged us to further investigate the efficacy of targeting BCL-2 and Onc-p53 for the treatment of SCLC patients.

## 3. Discussion

We have previously demonstrated that a subset of SCLC cells that express high levels of BCL-2 are sensitive to venetoclax monotherapy in in vitro and in vivo mouse models of SCLC, including PDXs [13]. Since then, leading experts in SCLC biology have proposed a classification of four SCLC subtypes [14]. We realized that several SCLC cell lines are venetoclax-resistant even though the levels of BCL-2 are high. These BCL-2-high but venetoclax-resistant cell lines all belong to either SCLC-A or SCLC-P subtypes in our analyses. In this regard, it has been demonstrated that ASCL1 binds to and activates the *BCL-2* promoter [29]. However, the mechanism by which POU2F3 activates the *BCL-2* promoter is not known and could be studied in the future. Furthermore, most of these venetoclax-resistant cell lines possess mutant p53. Therefore, we tested whether inactivation of mutant p53 could re-sensitize to venetoclax. For this purpose, we genetically inactivated mutant p53 by shRNA and pharmacologically by ganetespib, an HSP90 inhibitor, to destabilize and degrade mutant p53.

Based on our results in this study, we propose the following model as the mechanism of ganetespib and venetoclax treatment in BCL-2-high SCLC cells (Figure 8): (A) The inhibition of HSP90 triggers Onc-p53 degradation, followed by BIM induction. (B) Induced BIM localizes at the mitochondria. However, the activity of BIM is suppressed by BCL-2. (C) Venetoclax inhibits the interaction of BIM:BCL-2, which results in BIM followed by BAX activation and apoptosis. It is still unclear whether Onc-p53 directly binds to and regulates the *BIM* gene promoter as a putative repressor. Further studies are needed for clarification. 

BH3 mimetics to inhibit BCL-2, such as AbbVie’s venetoclax [11,30] and Ascentage’s APG-2575 (lisaftoclax) [31,32], are important new cancer therapeutics. Indeed, venetoclax has achieved FDA approval and has revolutionized treatment for hematological cancers that are reliant on BCL-2. The marked increases in patient responses to venetoclax compared to navitoclax across BCL-2-dependent malignancies—for instance, 79% for venetoclax [12] and 35% for navitoclax [33] in CLL—underline that navitoclax, limited by thrombocytopenia, was insufficient to robustly inhibit BCL-2 in patients. Additionally, the favorable toxicity profile of venetoclax, even in elderly populations, has led to over 100 clinical trials exploring venetoclax as part of combination therapies [34]. In contrast, the only solid tumors with noticeable sensitivity to venetoclax were SCLC and *MYCN*-amplified neuroblastoma [13,35,36], and there is an on-going clinical trial with venetoclax for neuroblastoma (NCT04029688). The current work suggests a rationale for clinical trials with the combination of venetoclax and HSP90 inhibitors in subsets of SCLC. 

HSP90 is a vital chaperone protein, regulating signaling pathways in cancer cells by interacting with oncogenic client proteins such as Onc-p53 [37,38]. The inhibition of HSP90 chaperone machinery has been demonstrated as a potential approach to inhibiting tumor cell survival, proliferation, invasion, and migration. Numerous HSP90 inhibitors have been developed as targeted therapies by suppressing the oncogenic pathways in cancer cells [39,40]. These inhibitors can be classified into: (i) N-terminal domain (NTD) inhibitors; (ii) C-terminal domain (CTD) inhibitors; and (iii) isoform-selective inhibitors. Ganetespib belongs to the NTD inhibitors and has been investigated in clinical trials, including SCLC. For example, a phase Ib/II study of ganetespib with doxorubicin in relapsed-refractory SCLC (NCT02261805) showed a 25% response rate with no dose-limiting toxicity [22]. However, the major limiting factors of HSP90 inhibitors are dose-limiting toxicity and poor pharmacokinetic profiles [41]. Novel HSP90-targeted compounds are constantly being discovered and tested alone or in combination in preclinical and clinical trials [42]. Thus, the combination of HSP90 inhibitors and venetoclax might be an alternative strategy for an effective cancer treatment.

Other than HSP90 inhibitors, Onc-p53 can be inactivated by inducing degradation or reactivating wild-type-like p53 activity [16]. A couple of reports have demonstrated that statins exclusively degrade mutant p53 by disrupting the SREBP-mevalonate pathway [43,44,45]. To restore the WT p53 function, APR-246 (PRIMA-1^Met^) causes proper mutant p53 refolding to WT p53 and has been tested in clinical trials [46]. Thus, it would be interesting to further examine whether other Onc-p53-targeting drugs would also show enhanced cytotoxicity in combination with venetoclax in Onc-p53-expressing SCLC cells. 

In conclusion, we rationally developed a venetoclax-based combination with the HSP90 inhibitor, ganetespib, that targets Onc-p53 in SCLC-A and SCLC-P subtypes. This strategy may stratify SCLC patients with biomarkers such as ASCL-1/POU2F3, BCL-2, and Onc-p53. 

## 4. Materials and Methods

### 4.1. Cell Lines and Drugs

H1048 (SCLC-P), H211 (SCLC-P), H146 (SCLC-A), H526 (SCLC-P), H345 (SCLC-A), and H209 (SCLC-A) cells were purchased from ATCC (Manassas, VA, USA). Cells were cultured in RPMI 1640 media (Thermo Fisher, 11875093, Waltham, MA, USA), supplemented with 10% heat-inactivated fetal bovine serum (FBS) (Sigma, St. Louis, MO, USA, 20K514), and 100 µg/mL penicillin G/streptomycin at 37 °C in a humified, 5% CO_2_ incubator. ABT-199 (MedChemExpress, HY-15531, Monmouth Junction, NJ, USA, and LC Laboratories, V-3579, Woburn, MA, USA) and Ganetespib (MedChemExpress, HY-15205, Monmouth Junction, NJ, USA) were dissolved in dimethyl sulfoxide (DMSO), and stable drugs were stored at −20 °C in the dark. The final concentration of DMSO was 0.1%.

### 4.2. Lentivirus Production

The lentiviral short-hairpin RNA (shRNA) constructs, shBIM and shp53, were purchased from Sigma and Thermo Fisher, respectively. Each shRNA plasmid was co-transfected into HEK293T cells with psPAX2 (Addgene, 12260, Watertown, MA, USA) and pMD2.G (Addgene, 12259, Watertown, MA, USA) using EndoFectin (GeneCopoeia, EF001, Rockville, MD, USA). Lentivirus-containing supernatants were collected and used to infect the cell line of interest, and stable cell lines were established by 2 µg/mL puromycin selection.

### 4.3. Cell Viability Assays

Cells were seeded at a density of 5 × 10^3^ cells/well in 96-well plates with 100 µL of RPMI media. Cells were treated with the indicated concentrations of either ABT-199, Ganetespib, or in combination for 72 h. Fifty µL of RPMI media and 3 µL of cell proliferation (WST-1) reagent (Sigma, 11644807001, St. Louis, MO, USA,) or equal amounts of RPMI and CellTiter-Glo 2.0 reagent (Promega, G9241, Madison, WI, USA) were added to the cells before analysis. The wavelength for measuring the absorbance of the formazan product was 450 nm. Analyses were performed using a microplate reader from Promega according to the manufacturer’s protocol. 

### 4.4. Western Blot Analyses

Cells were seeded at a density of 1.0 × 10^6^ in 10 cm dishes. Cells were treated with either vehicle (DMSO), 1 µM ABT-199, or 50 nM Ganetespib for the indicated times. Whole cell lysates were prepared using CHAPS buffer [20 mM Tris (pH 7.4), 137 mM NaCl, 1 mM dithiothreitol (DTT), 1% CHAPS (3-[(3-cholamidopropyl) dimethylammonio] 1-propanesulfonate)]. Equal amounts of protein were loaded into an SDS-polyacrylamide gel, transferred onto a nitrocellulose membrane, incubated with antibodies of interest, and analyzed with ECL2 western blotting substrate (Thermo Fisher, 32132, Waltham, MA, USA). Primary antibodies were used in a 1:1000 dilution for BIM (Cell Signaling, C34C5, Danvers, MA, USA), BAK (Cell Signaling, D4E4, Danvers, MA, USA), BCL-X_L_ (Cell Signaling, Danvers, MA, USA, 54H6), GAPDH (Cell Signaling, Danvers, D16H11, MA, USA), Cleaved-Caspase 3 (Cell Signaling, Danvers, MA, USA, D175), Cleaved-PARP (Cell Signaling, D64E10, Danvers, MA, USA), Human BCL-2 (Santa Cruz Biotech, sc-509, Dallas, TX, USA), MCL-1 (ENZO, ADI-AAP-240-F, Farmingdale, NY, USA), p53 (Santa Cruz Biotech, DO-1, Dallas, TX, USA), and BAX (Cell Signaling, D2E11, Danvers, MA, USA). Secondary antibodies, HRP-linked anti-rabbit IgG (Cell Signaling, Danvers, MA, USA) and HRP-linked anti-mouse IgG (Cell Signaling, Danvers, MA, USA), were used in a 1:2000 dilution.

### 4.5. Immunoprecipitation

Human BCL-2 (Santa Cruz Biotech, sc-509, Dallas, TX, USA), BIM (Cell Signaling, Danvers, MA, USA, C34C5), BAX (Santa Cruz Biotech, 6A7, Dallas, TX, USA), or rabbit/mouse IgG (primary antibodies, 1:100 dilution) were added to 500 µg of whole cell lysates and incubated with rotation at 4 °C overnight. Antibody complexes were then captured using Protein A/G UltraLink Resin (Thermo Fisher, 53132, Waltham, MA, USA) at 4 °C with rotation for 1 h. Samples were centrifuged, washed with CHAPS buffer 3×, and resuspended in CHAPS buffer and 5X SDS loading buffer. After boiling the samples for 5 min, they were analyzed by western blotting as described above.

### 4.6. Fluorescence-Activated Cell Sorting (FACS) Analysis

Cells were seeded at a density of 5.0 × 10^5^ cells in 60 mm dishes and treated with either vehicle, 1 µM ABT-199, 50 nM Ganetespib, or in combination for 24 h. Following treatment, cells were harvested, washed with PBS, and resuspended with 100 µL of 1× Annexin V binding buffer (BD Biosciences, 556454, Franklin Lakes, NJ, USA). Annexin-V-FITC (BioLegend, 640945, San Diego, CA, USA) and propidium iodide (PI) (Thermo Fisher, P3566, Waltham, MA, USA) were added, and cells were incubated in the dark for 15 min at room temperature. Then, 400 µL of 1× binding buffer were added to the suspension for analysis. Cells were analyzed using FACSCAN (BD FACSCanto) at the VCU Flow Cytometry Core and quantified the population as either double-negative, Annexin V-positive, PI-positive, or double-positive by the FlowJo software version 10.8.1.

### 4.7. Quantitative Reverse Transcriptase Polymerase Chain Reaction (qRT-PCR)

Cells were seeded at a density of 1.0 × 10^6^ in 60 mm dishes and treated with either vehicle, 50 nM or 100 nM Ganetespib for 24 h. Cells were harvested, and total RNA was extracted using Quick-RNA MiniPrep (Zymo, R1054, Irvine, CA, USA) following the manufacturer’s instructions. cDNA was synthesized using the Applied Biosystems High Capacity cDNA reverse transcription kit (Thermo Fisher, 4368814, Waltham, MA, USA) based on the manufacturer’s protocol. cDNA was amplified in triplicate using PowerTrack SYBR green master mix (Thermo Fisher, A46109, Waltham, MA, USA) in the StepOnePlus Real-time PCR system by Applied Biosystems. Primers were synthesized with the following sequence: GAPDH-Forward [5’-GTC TCC TCT GAC TTC AAC AGC G-3’] and GAPDH-Reverse [5’-ACC ACC CTG TTG CTG TAG CCA A-3’]. BIM-Forward [5’-CAA GAG TTG CGG CGT ATT GGA G-3’] and BIM-Reverse [5’-ACA CCA GGC GGA CAA TGT AAC G-3’]. mRNA expression was determined using ΔΔCt.

### 4.8. In Vivo Studies

All animal studies were conducted in accordance with the VCU Institutional Animal Care and Use Committee (IACUC) guidelines. Mice were purchased from the VCU Cancer Mouse Models Core. H1048 cells (2.0 × 10^6^) were subcutaneously inoculated at the flank of NOD-SCID-IL2gamma receptor null (NSG) mice (male; 6-week old). Once tumors became pulpable (~100 mm^3^), mice were randomized into four groups (7 mice/group) and treated with vehicle, ABT-199 (80 mg/kg) via oral gavage 3× a week, Ganetespib (100 mg/kg) via tail vein injection 1× a week, and in combination for 2 weeks. Tumor volumes were measured by calipers. Tumor volume was calculated as V = ½ × AB^2^, where A is the longest dimension of the tumor and B is the dimension of the tumor perpendicular to A. 

### 4.9. Statistical Analyses

All quantitative data are shown as ±S.D from at least two independent experiments, which were conducted in duplicate. Microsoft Excel was used for statistical analysis. All data were analyzed with paired and/or unpaired *t*-tests. All reported *p*-values were 2-tailed, and *p* ≤ 0.05 was considered statistically significant.

## Figures and Tables

**Figure 1 ijms-24-13082-f001:**
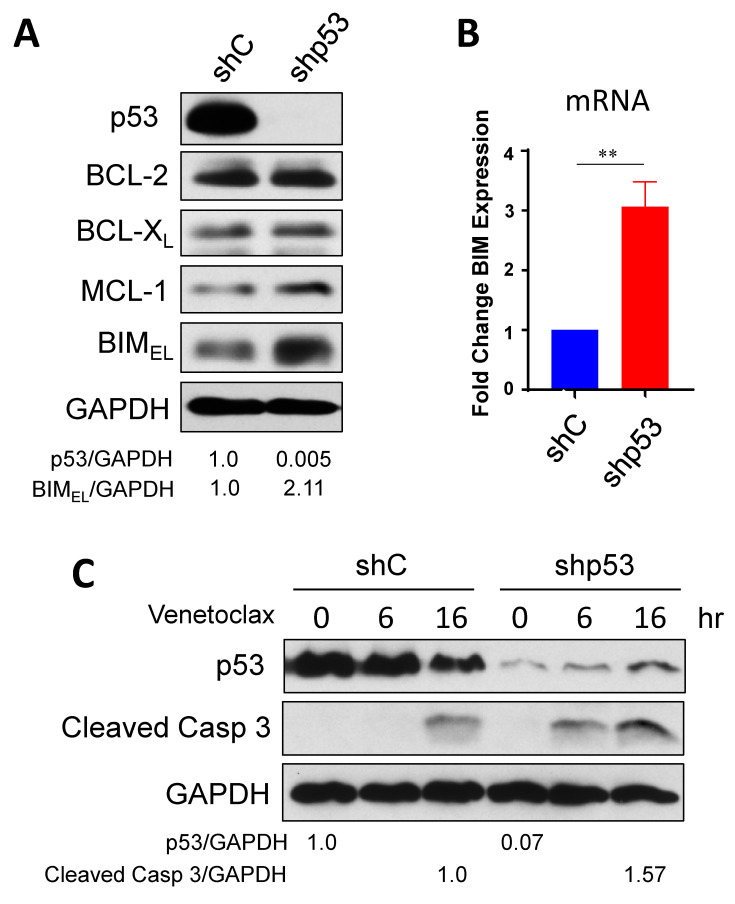
Down-regulation of Onc-p53 induces BIM and sensitizes SCLC to venetoclax. (**A**) Onc-p53 was downregulated by shRNA (shp53) in H1048 cells. The levels of BCL-2 family proteins compared to scrambled shRNA control (shC) were determined by Western blots with the indicated antibodies. (**B**) The levels of *BIM* mRNA in H1048 shp53 and scrambled RNA control cells were determined by qRT-PCR. (**C**) H1048 shp53 and scrambled RNA control cells were treated with 1 µM venetoclax. Total cell extracts were prepared at the indicated time points and subjected to Western blots with the indicated antibodies. ** *p* < 0.01. Values represent the mean ± S.D. of triplicates.

**Figure 2 ijms-24-13082-f002:**
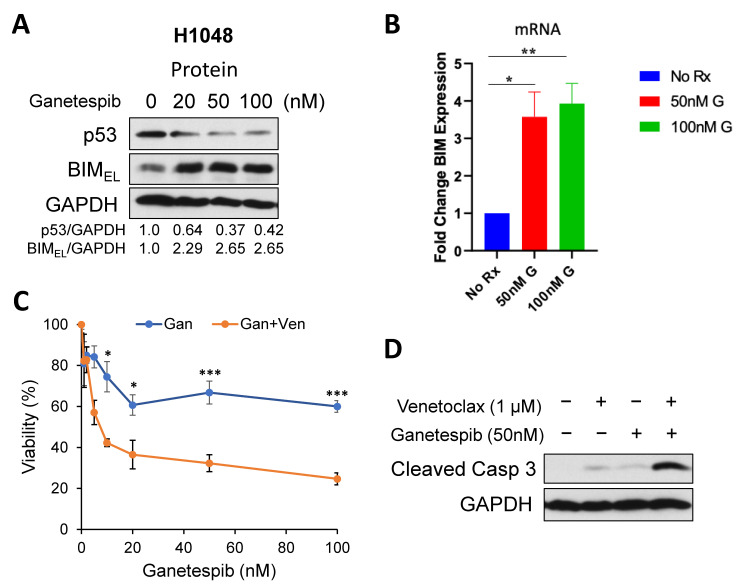
An HSP90 inhibitor, ganetespib, down-regulates Onc-p53, induces BIM, and enhances cell death in combination with venetoclax. (**A**) H1048 cells were treated with the indicated concentrations of ganetespib for 24 h. Total cell extracts were subjected to Western blots. (**B**) H1048 cells were treated with the indicated concentrations of ganetespib (G) for 24 h. Total RNA was subjected to qRT-PCR to determine the levels of *Bim* mRNA. (**C**) H1048 cells were treated with 1 µM venetoclax and indicated concentrations of ganetespib for 72 h. Cell viability was determined by the WST-1 assay. * *p* < 0.05, ** *p* < 0.01, *** *p* < 0.005. Values represent the mean ± S.D. of triplicates. (**D**) H1048 cells were treated with 1 µM venetoclax and/or 50 nM ganetespib for 72 h. Total cell extracts were subjected to Western blots with the indicated antibodies.

**Figure 3 ijms-24-13082-f003:**
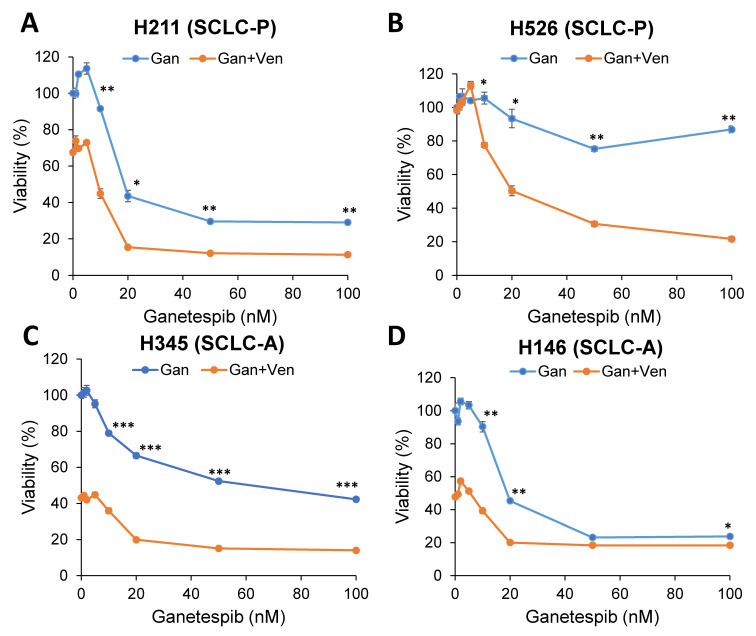
The effect of the venetoclax and ganetespib combination in venetoclax-resistant SCLC-A and SCLC-P cell lines. (**A**) H211, (**B**) H526, (**C**) H345, and (**D**) H146 cells were treated with 1 µM venetoclax and indicated concentrations of ganetespib for 72 h. Cell viability was determined by the WST-1 assay. * *p* < 0.05, ** *p* < 0.01, *** *p* < 0.005. Values represent the mean ± S.D. of triplicates.

**Figure 4 ijms-24-13082-f004:**
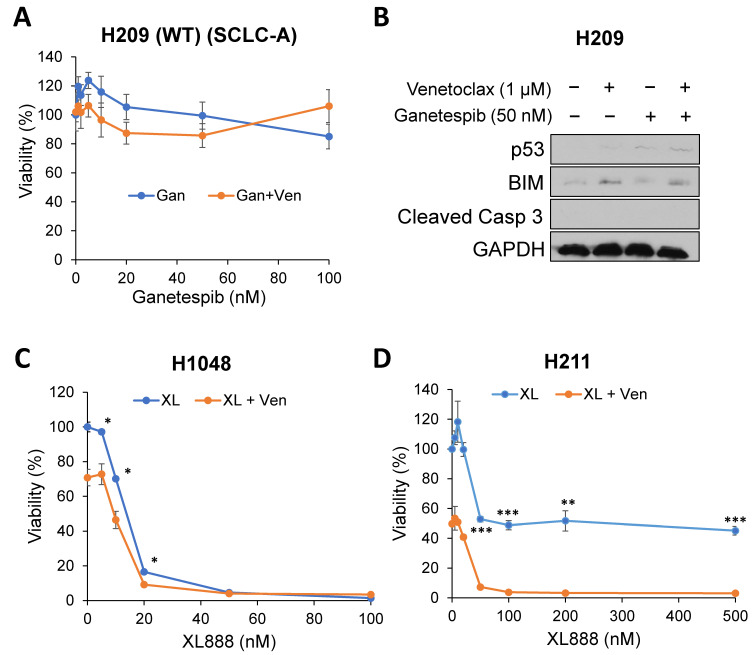
The effect of venetoclax and HSP90 inhibitor combinations in venetoclax-resistant SCLC-A and SCLC-P cell lines. (**A**) H209 cells were treated with 1 µM venetoclax and indicated concentrations of ganetespib for 72 h. Cell viability was determined by the WST-1 assay. (**B**) H209 cells were treated with 1 µM venetoclax and/or 50 nM ganetespib for 72 h. Total cell extracts were subjected to Western blots with the indicated antibodies. (**C**) H1048 and (**D**) H211 cells were treated with 1 µM venetoclax and indicated concentrations of XL888 for 72 h. Cell viability was determined by the CellTiter-Glo assay. * *p* < 0.05, ** *p* < 0.01, *** *p* < 0.005. Values represent the mean ± S.D. of triplicates.

**Figure 5 ijms-24-13082-f005:**
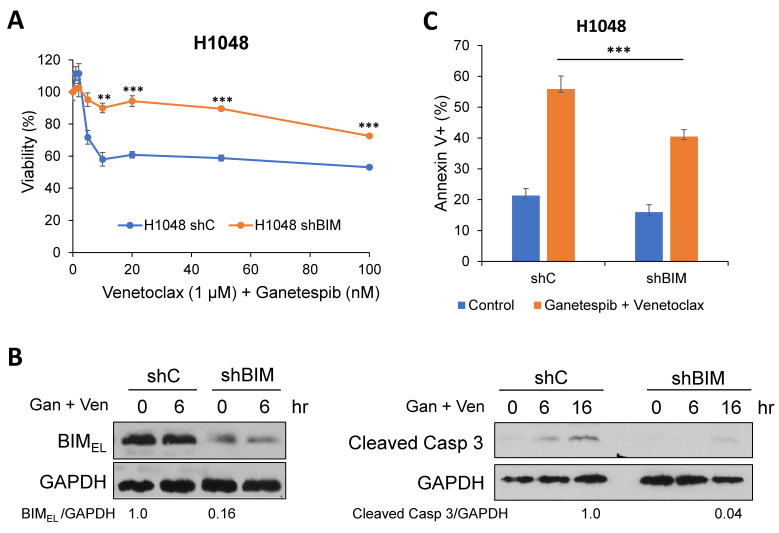
Down-regulation of BIM mitigates cell death induced by venetoclax and ganetespib. (**A**) BIM was downregulated by shRNA in H1048 cells. Cells were treated for 72 h with 1 µM venetoclax and indicated concentrations of ganetespib. Cell viability was determined by the WST-1 assay. Values represent the mean ± S.D. of triplicates. (**B**) H1048 shBIM and scrambled RNA control cells were treated with 1 µM venetoclax and/or 50 nM ganetespib. Total cell extracts were prepared at the indicated time points and subjected to Western blots with the indicated antibodies. (**C**) H1048 shBIM and scrambled RNA control cells were treated with 1 µM venetoclax and 50 nM ganetespib for 48 h. Apoptosis was determined by Annexin V-PI staining followed by FACS. ** *p* < 0.01, *** *p* < 0.005.

**Figure 6 ijms-24-13082-f006:**
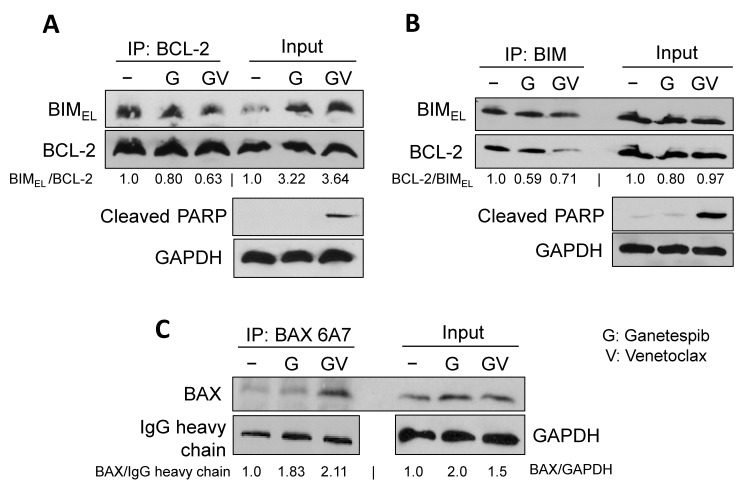
The BIM:BCL-2 complex was dissociated, and the effector BAX was activated by venetoclax treatment. H1048 cells were treated with 1 µM venetoclax and/or 50 nM ganetespib for 8 h. Total cell lysates were subjected to immunoprecipitation with (**A**) anti-BCL-2, (**B**) anti-BIM, or (**C**) anti-BAX 6A7 antibodies. Western blot analyses were carried out on precipitated samples with the indicated antibodies. Input demonstrated the integrity of each cell lysate.

**Figure 7 ijms-24-13082-f007:**
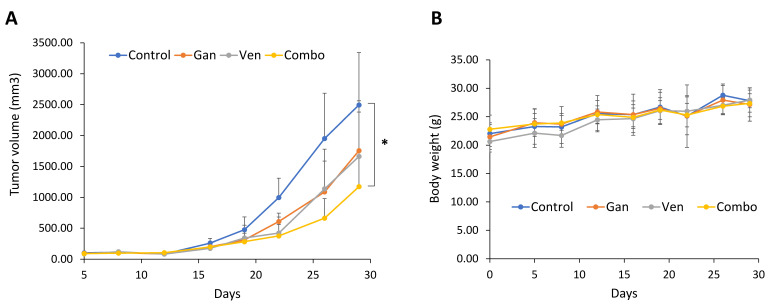
The combination of venetoclax and ganetespib delays tumor growth in a xenograft mouse model of SCLC. H1048 cells were subcutaneously inoculated in 6-week-old male NSG mice at the flank (Day 0). When tumors achieved a size of ~100 mm^3^, mice were randomized into four groups (Day 5, N = 7/group). Mice were treated with ganetespib (100 mg/kg) at Days 5 and 12 and/or venetoclax (80 mg/kg) at Days 6–9 and 13–16. (**A**) Tumor volume was determined by caliper measurements. (**B**) Body weight was measured at the same time as tumor volume. * *p* < 0.05. Error bars equal S.D.

**Figure 8 ijms-24-13082-f008:**
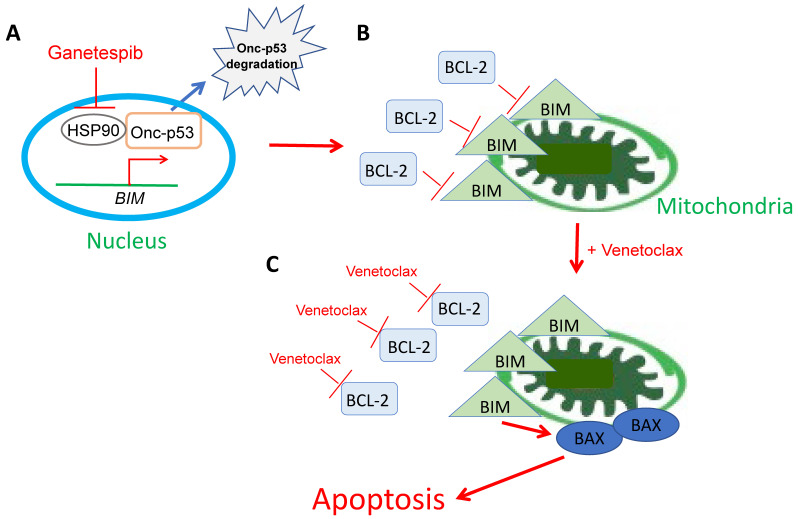
The mechanism of cell death induced by venetoclax and ganetespib in BCL-2-high SCLC cells. See details in the text above.

## Data Availability

All original data and images are contained within the article.

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
