# Peer review of "Targeting Oncogenic Mutant p53 and BCL-2 for Small Cell Lung Cancer Treatment"

_ijms, 2023, doi:10.3390/ijms241713082_

Round 1

Reviewer 1 Report

o Figure 2C shows that the IC50 of Gan after co-incubation with Ven was about 10 nM, but 50 nM of Gan was used in the following WB study (Figure 2D). Based on the curves in Figure 2C, cell survival was approximately 35% under the treatment of 50 nM of Gan and Ven. Is such a high concentration doable? Is it too toxic? What is the logic behind choosing this concentration for mechanism study? In contrast, this situation is not found in Figures 4 and 5. Please explain.

o The GAPDH bands are overexposed. I fully understand that this is due to the lower amount of target protein. Does this affect quantification? Do the authors have a photo with a shorter exposure time? It would be better to show another photo with a shorter exposure time.

o Figure 7, it would be better if the authors could add an image of the excised tumor tissue. This is more visual.

o In Section 4.8, there should be more detailed information about the mice, such as vendor, sex, age, weight, etc. 

o It would be better if the authors can describe in one or two lines why a particular assay is done and what they are looking for. Although they have mentioned the purpose of some studies but not for all the parts.

Author Response

  1. Figure 2C shows that the IC50 of Gan after co-incubation with Ven was about 10 nM, but 50 nM of Gan was used in the following WB study (Figure 2D). Based on the curves in Figure 2C, cell survival was approximately 35% under the treatment of 50 nM of Gan and Ven. Is such a high concentration doable? Is it too toxic? What is the logic behind choosing this concentration for mechanism study? In contrast, this situation is not found in Figures 4 and 5. Please explain.

Response: Thank you for these important comments. While the cell survival at 50 nM of ganetespib in combination with venetoclax was ~35% (Figure 2C), the single-dose treatment of ganetespib at 50nM was still above 60%, showing that this drug alone is not enough to be toxic to cells, but high enough to observe the oncogenic-p53 (Onc-p53) degradation as shown in Figure 2A. This Onc-p53 degradation further sensitizes previously resistant cells to venetoclax enough to dissociate the BIM:BCL-2 complex, activating BIM and BAX, and inducing cell death (Figure 6).

In Figure 4A, H209 cells are resistant to the combination treatment because p53 is wild-type. In Figure 5, the combination treatment did not result in a lower cell survival because BIM knockdown by shRNA prevents the activation of BIM, therefore inhibiting apoptosis from occurring.

We hope that this explanation is satisfactory to the reviewer.

  1. The GAPDH bands are overexposed. I fully understand that this is due to the lower amount of target protein. Does this affect quantification? Do the authors have a photo with a shorter exposure time? It would be better to show another photo with a shorter exposure time.

Response: The reviewer’s suggestion is highly appreciated, and we have updated the GAPDH bands with shorter exposures in Figure 5B and Figure 6B. The densitometry quantification of the proteins was also updated accordingly. However, the conclusion of these experiments has not been changed. We hope that these updates are satisfactory to the reviewer’s suggestion.

  1. Figure 7, it would be better if the authors could add an image of the excised tumor tissue. This is more visual.

Response: We thank the reviewer for this suggestion, but we believe that the caliper measurements of these tumors in Figure 7A show a more precise result of tumor growth delay compared to an image of excised tumor tissue. We don’t believe that the image of an unstained tumor tissue will provide a better understanding to readers. We hope the reviewer understands why we did not incorporate this into the manuscript.

  1. In Section 4.8, there should be more detailed information about the mice, such as vendor, sex, age, weight, etc. 

Response: We apologize for not including this important information. We have modified section 4.8 of the Materials and Methods section (line 367 and 369) to include the vendor, sex, and age of the mice. The weight of the mice for the entire study is found in Figure 7B. We thank the reviewer for these comments.

  1. It would be better if the authors can describe in one or two lines why a particular assay is done and what they are looking for. Although they have mentioned the purpose of some studies but not for all the parts.

Response: We agree with the reviewer that further explanation could have been included for particular assays. We have updated the description of Figure 6 (line 187-189) to include more detail about the mechanism of action of venetoclax and why we are performing the co-immunoprecipitation as follows:

Venetoclax is known to bind to the hydrophobic pocket of BCL-2 to inhibit the formation of BIM:BCL-2 complex followed by the activation of BAX, the downstream effector of BIM, to induce apoptosis.

Reviewer 2 Report

The authors have shown that the down-regulation of Onc-p53 increases the expression of a 24 BH3-only pro-apoptotic BIM and sensitizes to venetoclax in SCLC-P cells. Targeting Onc-p53 by the HSP90 inhibitor, ganetespib increases BIM expression and sensitizes to venetoclax in SCLC-P  and SCLC-A cells. This study provides evidence that targeting BCL-2 and Onc-p53 by the combination of venetoclax and  HSP90 inhibitors would be a promising approach for SCLC treatment. Altogether, this is an interesting study with novel findings. However, the study has many analytical flaws. Most of the experiments missing the p-value along with there are the following issues that can be taken care of before publication.

1. How ABT-199 and  Ganetespib reconstituted. If DMSO was used what was the final concentration of DMSO in the final treatment volume?

2. Figures 2 B and C, do not show  p-value, please provide it in the respective  panel.

3. Figure 3(A) H211, (B) H526, (C) H345 and (D) H146, p-value in missing in all panel. Control vs treated cells.

4. How was the percentage of cell viability calculated, please describe in the method section.

5. In Immunoprecipitation experiments, IP with normal IG control is missing. This does not rule  that venetoclax and ganetespib mediated association is specific or non-specific. Please explain it

6. In in vivo experiments authors are advised to present the status of p53, caspases, and BIM and BCL2. This will provide a very strong statement of the study. 

Author Response

  1. How ABT-199 and Ganetespib reconstituted. If DMSO was used what was the final concentration of DMSO in the final treatment volume?

Response: Reconstitution of ABT-199 and ganetespib consisted of DMSO, where the final concentration for in vitro assays was 0.1%. We have modified section 4.1 of the Materials and Methods (line 293) as follows:

The final concentration of DMSO was 0.1%.

  1. Figures 2B and C, do not show p-value, please provide it in the respective panel.
  2. Figure 3(A) H211, (B) H526, (C) H345 and (D) H146, p-value in missing in all panel. Control vs treated cells.

Response: We apologize for this missing information. We have now included p-value for these panels and modified the figure legends appropriately to include these values.

  1. How was the percentage of cell viability calculated, please describe in the method section.

Response: We agree with the reviewer that we may have left out a piece of information pertaining to how the cell viability is calculated. Analyses were performed based on the manufacturer’s protocol of measuring the absorbance of samples against background controls. To further clarify, we have edited this method to include the wavelength at which the samples were measured (line 315) as follows:

The wavelength for measuring absorbance of the formazan product was 450 nm

  1. In Immunoprecipitation experiments, IP with normal IG control is missing. This does not rule that venetoclax and ganetespib mediated association is specific or non-specific. Please explain it.

Response: We now show co-immunoprecipitation with normal IgG in the Supplementary Figure 3. These controls show minimum background and indicate that the IPs with anti-BIM or anti-BCL-2 antibodies are specific.

  1. In in vivo experiments authors are advised to present the status of p53, caspases, and BIM and BCL2. This will provide a very strong statement of the study. 

Response: We appreciate the reviewer’s concern relating to the importance of presenting the status of these proteins from an in vivo experiment; however, tumors can exhibit more heterogenicity compared to in vitro cell based assays. Therefore, the expression study in tumors could show unclear results compared to the in vitro study, which may cause confusion amongst readers. Furthermore, these pharmacodynamic studies need separate experimental design including timing of tumor collection and the methods to be analyzed (e.g., immunohistochemistry, protein extraction followed by Western blotting). Thus, we don’t think these analyses would help our main conclusion demonstrated in the current manuscript. 

Reviewer 3 Report

A very elegant and well carried out preclinical study in small cell lung cancer cell lines/SCLC-A and SCLC-P). The research work is well written and flawless. 

There are no caveats, perhaps the authors could bear in mind that transcription factors could also be causing resistance to the BCL2 inhibitor venetoclax including bromodomain containing protein 4 (BRD4). The protein translational inhibitor homoharringtonine (omacetoxine) could repress the enhancer BRD4 occupancy. Cotreatment with omacetaxine and venetoclax or a BET inhibitor has demonstrated in vitro synergy and improved antileukemic effects in vivo (Mill et al. Effective therapy for AML with RUNX1 mutation by cotreatment with inhibitor of protein translation and BCL2. Blood 2022.

Author Response

We appreciate the reviewer’s suggestion to look further into transcription factors that could be causing resistance to venetoclax.

Round 2

Reviewer 1 Report

Based on the current version and the response letter, all the suggestions and comments have been addressed. I recommend acceptance of the current version of the manuscript. 

Reviewer 2 Report

All queries were responded well.